# Entropy Generation of Forced Convection during Melting of Ice Slurry

**DOI:** 10.3390/e21050514

**Published:** 2019-05-21

**Authors:** Beata Niezgoda-Żelasko

**Affiliations:** Institute for Thermal and Process Engineering Cracow University of Technology, al. Jana Pawła II 37, 31-864 Kraków, Poland; bniezgo@mech.pk.edu.pl; Tel.: +48-12-628-3588

**Keywords:** ice slurry, entropy generation rate, melting process, forced convection, tube flow, heat exchangers, non-Newtonian fluid

## Abstract

This paper looks at entropy generation during ice slurry flow in straight pipes and typical heat exchanger structures used in refrigeration and air-conditioning technology. A dimensionless relationship was proposed to determine the interdependency between flow velocity and the volume fraction of ice, for which the entropy generation rates were at the minimum level in the case of non-adiabatic ice slurry flow. For pipe flow, the correlation between the minimum entropy generation rate and the overall enhancement efficiency was analyzed. As regards heat exchange processes in heat exchangers, the authors analyzed the relationship between the minimum entropy generation rate and the heat exchange surface area and exchanger efficiency.

## 1. Introduction

Ice slurry-based refrigeration is among the most state-of-the-art refrigeration technologies in indirect installations with and without cold storage. It is also used as a process supporting cold distribution in ice harvester systems. Ice slurry is a mixture of water ice crystals and water or water and a freezing-point depressant (salt, glycol, alcohol, etc.). The ice particle diameters most often amount to 10^−4^–0.5 mm. The mass fraction of ice crystals in the ice slurry, which may be transported in practice, do not exceed 30%. As an environmentally neutral and natural heat carrier, ice slurry demonstrates a refrigeration potential commensurate to that of refrigerants. Ice slurry flow may be accompanied by phase segregation, which leads to a change in the mean density values and in the value of the dynamic coefficient of fluid viscosity, consequently leading to higher flow resistance values [1]. The heat transfer process in the ice slurry is accompanied by micro-convection of solid particles, resulting in an increase in thermal conduction coefficients and an enhancement of the heat transfer process. Ice slurries differ significantly from single-phase heat carriers in terms of their rheological properties. Ice slurries are non-Newtonian fluids [2,3,4,5,6,7,8,9,10]. The phenomena associated with flow resistance and heat transfer processes involving ice slurries have been extensively discussed in articles, such as [2,3,4,5,6,7,8,9,11,12]. A significant operational problem of ice slurry-fed systems is the elimination of phase segregation and ensuring a homogenous flow. According to [13,14], regardless of the ice content, homogenous flow of ethanol-based ice slurry is possible at a minimum flow velocity of 0.54 ms^−1^ [13] and 0.75 ms^−1^ [14] for 0.02 and 0.024 m diameter pipes, respectively. Flow velocity cut-off values for homogenous and heterogeneous flow in d = 0.02 m pipes for 10.5% ethanol-based ice slurry, calculated in [15] on the basis of a solid particle distribution model according to Kitanowski [4,16,17,18], for a 5% and 25% volume fraction of ice amount to 0.68 ms^−1^ and 0.15 ms^−1^, respectively.

While dynamic effective viscosity coefficients of slurries are higher than dynamic viscosity coefficients of carrier fluids (and they increase as the mass fraction of ice grows), the slurry flow resistance values are not always higher than those of carrier fluids at the same flow velocities. For an ice content of *x_s_* ≥ 10% and the same flow velocity, ice slurry flow resistances may be smaller than the flow resistances of the carrier fluid, provided that the type of flow of the carrier fluid and of the ice slurry is different [3,19,20,21]. In the case of homogeneous flow, these phenomena can be explained by the effect of the absorption of turbulence kinetic energy by ice particles (flow laminarization) and the shifting of the moment of transition to the turbulent flow area by slurries with high mass fractions of ice and higher Reynolds number than in the case of the carrier liquid. For heterogeneous flow, which is favoured by low ice mass fractions and larger dimensions of solid particles, it is difficult to interact with the carrier fluid and solid particles that accumulate in the top part of the tube. This effect favors an increase in turbulence of the carrier liquid and an increase in flow resistance. The character of motion changes at lower speeds and lower Reynolds numbers.

Heat exchange processes involving ice slurries are determined by several phenomena. Solid particle micro-convection associated with moving ice slurries significantly improves the thermal conduction coefficient values with respect to those determined for non-moving slurries, for example, using the Maxwell–Tareff equation [22,23]. An analysis of the Charunyakorn [24] equation implies that higher fluid shear rates correspond to higher thermal conduction coefficient values. The ice melting process is characterized by significantly higher specific thermal efficiency as compared to heat exchange during single-phase flow. These phenomena make ice slurry heat transfer coefficients higher than those of single-phase refrigerants. For example, for ice slurries based on an aquatic solution of ethanol, in the laminar flow area, the presence of solid particles leads to a 2–5.7-fold increase in the heat transfer coefficient with respect to the heat transfer coefficients of the carrier fluid (ethanol *x_ai_* = 10.6% at −4.5 °C) [2,12,25]. On the other hand, there are some velocities at which for ice slurries with an ice content of *x_s_* > 10%–20%, the heat transfer coefficients are lower than those of the carrier fluid. In the turbulent flow area, the ice content is not as significant in terms of its impact on the heat transfer coefficient value, although the values of these coefficients are about 20%–30% higher than the heat transfer coefficients of the carrier fluids [2,12,25].

Considering the maximization of the heat transfer coefficients and the minimization of flow resistance values, for ice slurry-fed installations, the *d_i_, w, x_s_* parameters should be selected so that the ice slurry flow is homogenous and laminar, with heat transfer coefficients exceeding those of the carrier fluid. Laminar flow favors lower flow resistances, which are always higher for ice slurries in the turbulent flow area than the flow resistances of the carrier fluid with the same flow velocity.

The large mass fraction of ice makes it possible to limit the mass flux of the refrigerant and the heat exchange surface area of ice slurry-fed heat exchangers. Without a doubt, ice particles enhance the heat transfer process. The efficiency of ice slurry use as a refrigerant can be determined on the basis of the criteria commonly used in heat transfer. The obvious indicators of the efficiency of various methods used to improve heat exchange conditions include the heat transfer coefficient enhancement ratio (ε*_h_* = *h/h_ref_*) and the flow resistance increase ratio ($\varepsilon_{\Delta p}=\Delta p/\Delta p_{ref}$). The mutual relationship between ε*_h_* and εΔ*_p_* is expressed as their quotient (ε*_h−__*Δ*p_* = ε*_h_*/εΔ*_p_*) [26], or the so-called overall enhancement efficiency, defined as ε*_Nu−f_ =* (Nu/Nu*_ref_*)/(*f/f_ref_*)^1/3^ [27,28]. The use of the second principle of thermodynamics in design is based on an analysis of the entropy generation rate, which enables a quantitative determination of the irreversibility of the analyzed thermal and flow process. It is common practice to use the entropy generation rate for the thermo-dynamic assessment of thermal and flow processes and as an optimization criterion, for example, in the design of heat exchangers. Also used are complex criteria for the assessment of efficiency—a combination of the performance evaluation criteria and the entropy generation minimization (EGM) criterion [29]. In [30], the authors propose a method for the analysis of the entropy generation rate in the design of single-phase thermal processes, heat storage systems or various thermal cycles. The method proposed by Bejan [30] has been successfully used by other authors to perform a thermodynamic assessment of thermal processes during single-phase flows [31,32], multi-phase flows [33,34,35,36,37,38], heat exchange involving non-Newtonian fluids [32,39,40] and design of thermal systems and exchangers [40,41,42,43,44,45,46,47,48].

In the boiling process, an analysis of the entropy generation rate was used, for example, to determine the preferred scope of use of plain and micro-finned tubes [36] or to optimize the geometrical parameters of the pipes of an R22 evaporator [37]. An analysis of the entropy generation rate makes it possible to determine the upper limit of the mass flux density used in micro-finned pipes. The authors of [36] demonstrated that the entropy generation minimization criterion cannot be the only optimization criterion in the design of heat exchangers and that multi-criterion optimization of heat exchanger geometry should be preferred. In this context, entropy generation and material cost minimization was used in [42] to design crossflow plate heat exchangers. The minimization of the dimensionless number of entropy generation made it possible to determine the optimum geometrical parameters for finned heat exchanger plates. Heat flux maximization and entropy generation minimization in the design of fluid coolers and air-cooled condensers was the topic discussed in [41]. The authors stated that while in single-phase heat exchangers entropy generation minimization leads to the maximization of thermal efficiency, this condition is not met in condensers.

Paper [49] is a more general publication about the usefulness of determining entropy generation rates in thermal and flow-related numerical calculations. Analyzing, among other things, the melting process, the authors identified a correlation between the correct determination of the entropy generation rate and numerical errors, convergence criteria and the selection of the time-step. In another study [45], the authors presented the use of the VEG (virtual entropy generation) method in measurement technique to reduce measurement errors in the studies of heat exchangers.

The issues concerning entropy generation in thermal and flow processes with ice slurry were discussed in the studies [38,39,40,46].

The entropy generation rate in the heat exchange process involving ice slurries was discussed in [38], where two types of aquatic slurries with PCM elements and multi-walled carbon nanotubes (MCNT) were compared. An analysis of the entropy generation rates found that for small pipe diameters and low heat flux densities, the use of PCM was preferred. Greater pipe diameters and higher heat flux densities, especially in the laminar flow area, preferred the use of aquatic slurries with the addition of MCNT. In the turbulent flow area, it is preferable to use water as the heat carrier, as it generates the lowest entropy generation rates. In another study [39], the analysis of entropy flux generation was limited to the case of laminar, isothermal flow of ice slurry which was an Ostwald–deWaele fluid. The authors of the paper [46] used the second principle of thermodynamics for the analysis of heat balance and entropy in a heat exchanger supplied with ice slurry of an aqueous solution of ethanol. In this paper, the effect of flow processes and heat transfer on the coolant and ambient side is not taken into account. The assumptions adopted by the authors implied the use of slurries with low mass fractions of ice. A comprehensive approach to the analysis of entropy flux in thermal processes with ice slurry has been presented in a previous publication [40]. The authors analyzed the entropy flux generated in a spiral heat exchanger supplied with ice slurry. It was indicated that it is possible to determine optimal Dean number values, for which the entropy flux is minimal, and the maximum thermal efficiency is achieved for specific mass fractions of ice.

An analysis of the above papers suggests a very broad use of entropy generation rates for the assessment of diverse thermal [32,35,38,43], and flow processes, in which flow parameters [31,33,34,36] and geometrical parameters may be optimized [37,41,42]. The determined entropy generation rates refer to single processes [36,38], or to devices where processes involving various agents take place [41,42,43,44]. Local or global values of the entropy generation rate may be determined analytically [32,33,34,35,38,39,42,44,46], or numerically [31,36,37,40,49,50].

The use of ice slurry in refrigeration processes is a heat transfer enhancement method applied in refrigeration and air-conditioning technology. Ice slurry fed into a heat exchanger is characterized by both mass flux and mass fraction of ice, which determine the heat transfer process intensity and momentum exchange, inducing various entropy generation rates. This article attempts to identify the range of velocity and mass fraction of ice values which ensure the minimum entropy generation rate during the flow of ethanol-based ice slurry, being a non-Newtonian Bingham’s fluid, through straight 0.01–0.02 m pipes. In addition, the study contains aconducted an analysis of the total entropy generation rate in lamelled air coolers and plate fluid coolers in the context of the optimization of the mass fractions of ice and mass fluxes of the ice slurry. In this case, the analysis focused on the relationships between ice slurry’s flow parameters, the heat exchange surface area, the heat exchanger’s efficiency and the total entropy generation rate on the side of the refrigerant and the cooled medium. New aspects presented in the paper refer to a comprehensive approach to thermodynamic analysis of thermal-flow processes with ice slurry, both for flow in straight tubes as well as in heat exchangers. The analysis of minimum total entropy fluxes in heat exchangers allowed to verify the usefulness of the results of the minimization of the entropy flux of ice slurry in the conditions of forced convection of ice slurry in straight tubes. Additionally, the study compares the results of determination of optimal parameters of flow velocity and mass fractions of ice according to the criterion of minimum entropy and the condition of ε*_Nu−f_* > 1. The innovative element presented in the paper was proposing a dimensionless relationship to identify the interdependency between the flow velocity and the mass fraction of ice for which the entropy generation rate was at its minimum level.

## 2. Entropy Generation Rate during Flow in a Straight Pipe

### 2.1. Entropy Generation during the Melting of Ice Slurry; Flow with Phase Separation

Assuming a homogenous single-dimensional flow of the ice slurry mass flux ${\dot{m}}_{IS}$ in a horizontal pipe (Figure 1), the first and second principle of thermodynamics for the dl test section may be described using Equations (1) and (2), respectively:(1)$$di_{IS}=T_{IS}\;ds_{IS}+\nu_{IS}\;dp_{IS}$$
(2)$$d{\dot{S}}_{l}\;dl=d({\dot{m}}_{S}\;s_{s}+{\dot{m}}_{F}\;s_{F})-\frac{d\dot{Q}}{T_{w}}$$

In the test section, the ice slurry entropy changes as per Equation (3):(3)$${\dot{m}}_{IS}\;ds_{IS}=d({\dot{m}}_{S}\;s_{s}+{\dot{m}}_{F}\;s_{F})$$

Using Equations (1) and (3), Equation (2) can be transformed into (4):(4)$$d{\dot{S}}_{l}\;=\frac{d\dot{Q}}{dl}\left(\frac{1}{T_{w}}-\frac{1}{T_{IS}}\right)+\frac{{\dot{m}}_{IS}\;}{T_{IS}}\nu_{IS}\;\left(-\frac{dp_{IS}}{dl}\right)$$

Writing the Fourier’s equation for the ice slurry heat transfer process (5) and using the relationship $d\dot{Q}=\dot{q}\;dA=\dot{q}\;P\;dl$, Equation (4) can be transformed into (6):(5)$$\Delta T=T_{w}-T_{IS}=\frac{\dot{q}}{\alpha_{IS}}$$
(6)$$d{\dot{S}}_{l}\;=\frac{{\dot{q}}^{2}P}{\alpha_{IS}\;T_{IS}^{2}+\dot{q}T_{IS}}+\frac{{\dot{m}}_{IS}\;}{T_{IS}}\nu_{IS}\;\left(-\frac{dp_{IS}}{dl}\right)=d{\dot{S}}_{l-H}+d{\dot{S}}_{l-\Delta p}$$

Equation (6) makes it possible to determine the entropy generation rate for a unit length of the pipe. Equation (6) is true for water-based ice slurry with a constant melting point. For ice slurries based, for example, on a 10.6% aquatic solution of ethanol, the temperature glide with complete melting of ice in a 30% ice slurry amounts to 2.2 K. Therefore, it was assumed that in (6), *T_IS_* corresponded to the mean temperature of the ice slurry, $T_{IS}={\bar{T}}_{IS}$.

### 2.2. Ice Slurry Flow and Heat Transfer in Pipes

The determination of the entropy generation rate from Equation (6) for non-adiabatic flow of ethanol-based ice slurry requires the identification of both the pressure drop and the heat transfer coefficients for specific heat transfer conditions. The ice slurry under consideration is a Bingham fluid [3,19] and its flow can be treated as a generalized flow of a non-Newtonian fluid. Equation (7) introduces the generalized form of the Reynolds number:(7)ReK=ρISw2−n*dhn*8n*−1K*

By using Equation (7), Fanning’s factors for technically plain pipes can be found for the laminar and turbulent flow area on the basis of Equations (8) and (9), respectively:(8)cf=16ReK
(9)cf=0.079ReK0.25

Ultimately, ice slurry flow resistance values are calculated using Equation (10).
(10)$$\Delta p=c_{f}\frac{2\;l\;\rho_{IS}\;w_{2}}{d_{h}}$$

Equation (7) makes it necessary to determine a characteristic flow-behavior index *n*^*^ and consistency index *K*^*^. For Bingham fluid and an arbitrary geometry of cross-section, these parameters are found from equations included in Table 1 [19].

For the generalized non-adiabatic flow of ice slurry, the heat transfer coefficient may be calculated for laminar and flow areas from Equations (11) and (12), respectively [25]:(11)$${\mathrm{Nu}}_{L}=B_{1}\;{\left({\mathrm{Gz}}_{K}\right)}^{B_{2}}\;{\left(\frac{\Delta x_{s}\cdot K_{F}}{100}\right)}^{B_{3}}{\left(\frac{d_{s}}{d_{h}}\right)}^{B_{4}}{\left(\frac{K^{*}\left(T_{IS}\right)}{K^{*}\left(T_{w}\right)}\right)}^{B_{5}}$$
(12)$${\mathrm{Nu}}_{T}=B_{1}\;{\left({\mathrm{Pe}}_{K}\right)}^{B_{2}}\;{\left(\frac{d_{s}}{d_{h}}\right)}^{B_{4}}$$

The constants *B_i_* in Equations (11) and (12) are shown in Table 2 [25]. Equations (7)–(12) are the result of own experimental studies carried out for ethanol ice slurry, for which measurement stations, the course of experiments, measurement accuracy and results were discussed in detail in studies [2,12,19,25,55]. The experimental scope of the applicability of Equations (7)–(12) is provided in Table 2.

### 2.3. Results of Calculations

The entropy generation rate was studied for forced convection during melting a 10.6% ethanol solution-based ice slurry with a mass fraction of ice of 0 <*x_s_* < 30%. Calculations were made for ice slurry flow through straight pipes with diameters *d_i_ =* 0.01, 0.016, 0.02 m, mass flux of ice slurry $0.04\leq {\dot{m}}_{IS}\leq 0.62$ kgs^−1^ and heat flux density $2\leq \dot{q}\leq 10$ kWm^−2^. Figure 2 presents the change in the total entropy generation rate as a function of the flow velocity and the mass share of ice during ice slurry flow through a *d_i_* = 0.016 m pipe with a constant heat flux density $\dot{q}=8$ kWm^−2^.

Figure 2 suggests that in the laminar flow area, the lowest entropy generation rates correspond to the maximum fraction of ice in the ice slurry. However, let it be noted that in the entire flow area, the lowest entropy generation rates correspond to low mass fractions of ice *x_s_* < 10% and the minimum value of the entropy generation rate is recorded in the turbulent flow area. This is related to the mutual relationship between entropy generation in thermal and flow processes (Figure 3).

In the studied range of flow velocities 0 ≤ *w* ≤ 3 ms^−1^ for an ice content greater than *x_s_* > 21.9%, the flow is laminar in nature, and the ice slurry flow resistance values are moderate, while the presence of solid particles favors the highest values of heat transfer coefficients [2,12]. In this case, entropy generation is determined by the irreversibility of the thermal processes. For low mass fractions of ice, obtaining suitably high values of heat transfer coefficients depends not only on the presence of solid particles, but also on the appropriate value of flow velocity. As a consequence, this leads to a change into the turbulent flow area, where, in turn, flow resistance values have a significant impact on the induction of the total entropy generation. It also needs to be noted that large mass fractions of ice (x_s_ > 20%) are typically associated with a relatively large range of velocities, for which the total entropy generation rate is at its minimum. This results from the dominant impact of solid particles and phase change, but not flow velocity, on heat transfer coefficients [2]. The calculations made for various heat flux densities and pipe diameters made it possible to determine for specific mass fraction of ice, the flow velocities for which the total entropy generation rate was at its minimum.

The analysis of Equations (6)–(12) and calculation results presented in Figure 2; Figure 3 indicate that in case of non-adiabatic ice slurry flow, the generated entropy flux is determined by flow velocity, mass fraction of ice, heat flux density and tube diameters. It should also be noted that the mass fraction of ice, flow velocity and diameter determine the nature of the flow of the agent. With the increase in the Reynolds number, the effects of particulate matter on the heat transfer coefficients decreases and a greater influence of flow phenomena on the generated entropy fluxes is observed. Therefore, in the turbulent flow area, the entropy flux depends more on flow velocity and diameter than mass fraction of ice and heat flux density. Small tube diameters imply higher values of heat transfer coefficients and low values of the entropy flux generated by the heat exchange process. In general, lower tube diameters correspond to lower total values of the entropy flux. The results presented in Figure 4a,b indicate that in the laminar flow area, the effect of tube diameter on entropy flux is particularly significant: $1.5\leq {\dot{S}}_{di}/{\dot{S}}_{dio}\leq 6.2$ for $1.6\leq d_{i}/d_{io}\leq 2$. In the turbulent flow area where the values of the Bejan numbers are high (Be > 0.5) the ${\dot{S}}_{di}/{\dot{S}}_{dio}$ ratio stabilizes at ${\dot{S}}_{di}/{\dot{S}}_{dio}$ = 1.34–1.4 and ${\dot{S}}_{di}/{\dot{S}}_{dio}$ = 2.7–3.7 for $d_{i}/d_{io}$ = 1.6 and $d_{i}/d_{io}$ = 2, respectively.

Figure 5 presents the relationship *w(x_s_)_S_*_min_ for various heat flux densities and the flow of ice slurry through a pipe with a diameter of *d_i_* = 0.016 m. The values of *w, x_s_*, determined analytically, were validated using experimental values.

The experimental values of *w* and *x_s_* were determined by calculating, at each measurement point, the entropy generation rate (Equation (6)) using the measured values of heat transfer coefficients and the flow resistance values [2,25]. Next, for the given values of *x_s_,* the measurement points were chosen at which $\dot{S}={\dot{S}}_{\min}$ and the non-monotonic course of the relationship *w(x_s_)_S_*_min_ was associated with an area of transition between laminar and turbulent flow. The plotted theoretical $w_{cal-\dot{S}\min}(X_{V})$ curves provide a qualitatively correct description of the course of analogous curves obtained directly from the measuring points. The mean relative difference between the experimental and analytical flow velocities $w_{\dot{S}\min}(X_{V})$ does not exceed 7.6%. The maximum difference between the measured velocities and the calculated velocities concerned the case of heat flux density $\dot{q}$ = 2 kW and was lower than 30%.

Figure 5 shows the critical values of *x_sC_*, and *w_C_* found on the basis of the criterion for motion nature change valid for the ice slurry (Equation (13)) [54]:(13)ReBC=10000 He (dsdi)0.251.25 He+334.9

All analyzed parameters (*x_s_, d_i_,*
*w*, $\dot{q}$) are independent, but they affect the heat transfer coefficients and flow resistance, and thus the entropy flux and its minimum value, in various ways. Charts similar to the one presented in Figure 5 allow the selection of parameters *x_s_, *(*X_v_*
Table 1) *w*, *d_i_*, $\dot{q}$ in such a way that the entropy flux generated by the ice slurry is minimal. The results presented in Figure 5 and similar results for tube diameters *d_i_* = 0.01 m and *d_i_* = 0.02 m can be presented in the form of criteria relationship (14). Correlation (14) defines the relationship between the Reynolds number and dimensionless numbers taking into account the effect of geometric (*K_qX_*) and flow parameters (*K_q_*, *K_qX_*) on the generated minimal entropy flux.
(14)ReIS−Smin=Kq+KqX
where
(15)$$K_{q}=C_{1}{\left(\frac{{\dot{q}}_{o}-\dot{q}}{{\dot{q}}_{o}}\right)}^{C_{2}}$$
(16)$$K_{qX}=\exp\left(C_{3}+C_{4}\ln\left(\frac{d_{i}}{d_{io}}\right)+C_{5}\frac{\dot{q}}{r{\dot{G}}_{o}}\left(\frac{X_{v}}{100}\right)\right)$$

The parameters *C_i_, d_io_*, ${\dot{q}}_{o}$, ${\dot{G}}_{o}$ are given in Table 3.

Equation (14) makes it possible to determine the optimum velocity values for the condition $\dot{S}={\dot{S}}_{\min}$, for the assumed mass fractions, heat flux densities and pipe diameters with the maximum relative error of 15% (mean error: 1.2%).

Figure 6 presents the relationship between the Reynolds number for the ice slurry (Re_IS_ calculated as for Bingham’s fluid) and $\dot{q}X_{v}/r$.

Figure 7 presents a comparison of the cut-off values of *x_s_, w* both for $\dot{S}={\dot{S}}_{\min}$ and ε*_Nu−f_* = (Nu/Nu*_pt_*)/(*f/f_pt_*)^1/3^ = 1, for ice slurry heat transfer during ice slurry flow through *d_i_* = 0.01 and 0.02 mpipes with constant heat flux density values of $\dot{q}$ = 5, 10 kWm^−2^. The range of *x_s_* and *w* values, for which ε*_Nu−f_* > 1, means that while the same power is required to transport the refrigerant, the use of ice slurry over the use of the carrier fluid as the refrigerant translates into better thermal efficiency. The results shown in Figure 7 suggest that the minimum entropy generation rate condition generally shifts the scope of ice slurry use towards higher flow velocities. The exception is low heat flux densities for the minimum pipe diameter under consideration, *d_i_* = 0.01 m, for which entropy generation minimization implies lower values of flow velocity with respect to the condition ε*_Nu−f_* > 1. Figure 2 and Figure 7 show that, regardless of the adopted criterion for the assessment of the heat exchange process involving ice slurry, the laminar flow range is especially preferred as regards the selection of *x_s_* and *w.* In design practice, the recommended flow velocities in pipes amount to *w* < 1 ms^−1^. This velocity range for the flow of ice slurry in pipes with *d_i_* < 0.02 m enables effective enhancement of the heat transfer process with the minimum entropy generation rates.

The calculations performed suggest that:○During pipe flow, the lowest entropy generation rates were characteristic of small mass fractions of ice in the turbulent flow area and for the flow velocity of 1.5 < *w* < 2 ms^−1^.○In the laminar flow area, the lowest entropy generation rates corresponded to the highest analyzed mass fraction of ice *x_s_* = 30%.○Regardless of the share of solid particles, the minimum entropy generation rate criterion requires the application of high flow velocities, which for heat flux density values of $\dot{q}\geq$ 10 kWm^−2^ are greater than *w* > 1 ms^−1^.

## 3. Entropy Generation in Heat Exchangers Fed with Ice Slurry

An analysis of the entropy generation rate in heat exchangers was performed under conditions which differ from those present during flow through pipes. For pipe flow, a specific heat flux density, flow velocity and mean mass share of ice are assumed for which the entropy generation rates are calculated. For heat exchangers, the exchanger’s thermal efficiency, ice slurry mass flux and the inlet value of the mass fraction of ice are assumed. A change in ice slurry enthalpy, the heat exchange surface (heat flux density) and the end temperature of the cooled medium result from the balance equations and, as a consequence, make it possible to determine the entropy generation rate.

The heat transfer process enhancement, greater specific enthalpy of the ice slurry and the almost constant melting point enable ice slurry-fed heat exchangers to have a smaller heat exchanger surface than heat exchangers which rely on single-phase refrigerants. The heat exchange surface area has an impact, for example, on the values of flow resistance in the heat exchanger. In this part of the article, the authors discussed the impact of the ice slurry mass flux and the mass fraction of ice on the change in the total entropy generation rate in heat exchangers. To achieve this, the heat transfer process was discussed for two types of heat exchangers: A lamelled air cooler and a plate heat exchanger, which was used as a milk cooler.

### 3.1. Air Cooler

Assuming that the melting process of the ice slurry occurs at a constant temperature, it is possible, on the basis of [56], to express the entropy generation rate in the ice slurry melting and air cooling process using Equation (17):(17)$$\Delta {\dot{S}}_{H}=\frac{{\dot{Q}}_{m}}{{\bar{T}}_{IS}}+{\dot{m}}_{a}\;c_{pa}\ln\left(\frac{T_{a-out}}{T_{a-in}}\right)$$

Taking into consideration the ice melting process in the ice slurry, Equation (17) may be converted into (18)
(18)$$\Delta {\dot{S}}_{H}=\frac{\dot{m}\;\Delta i_{IS}}{{\bar{T}}_{IS}}+{\dot{m}}_{a}\;c_{pa}\ln\left(\frac{T_{a-out}}{T_{a-in}}\right)$$

If the fluid flow within the heat exchanger is treated as quasi-adiabatic flow, we can state that
(19)$$T\;d{\dot{S}}_{\Delta p}=-\dot{V}\;dp$$
Whence, for air treated as an ideal gas ($p\dot{V}=\dot{m}R\;T$), Equation (19) can be transformed into (20)
(20)$$d{\dot{S}}_{\Delta pa}=-{\dot{m}}_{a}\;R\frac{dp}{p}$$

Equation (20), following integration, leads to Equation (21):(21)$$\Delta {\dot{S}}_{\Delta pa}=-{\dot{m}}_{a}\;R\;\ln\left(\frac{p_{a-out}}{p_{a-in}}\right)$$
Disregarding the change in volume of the ice slurry in the melting process, you can transform Equation (19) for a liquid refrigerant into (22)
(22)$$\Delta {\dot{S}}_{\Delta p-IS}=-{\bar{\dot{V}}}_{IS}\frac{\Delta p}{{\bar{T}}_{IS}}$$
Hence, the total entropy generation rate in the ice slurry-fed air cooler may be calculated from Equation (23):(23)$$\Delta \dot{S}=\Delta {\dot{S}}_{H}+\Delta {\dot{S}}_{\Delta p}=\frac{\dot{m}\;\Delta i_{IS}}{{\bar{T}}_{IS}}+{\dot{m}}_{a}\;c_{pa}\ln\left(\frac{T_{a-out}}{T_{a-in}}\right)-{\dot{m}}_{a}\;R\;\ln\left(\frac{p_{a-out}}{p_{a-in}}\right)\;-{\bar{\dot{V}}}_{IS}\frac{\Delta p}{{\bar{T}}_{IS}}$$
When determining the value of the entropy generation rate for an air cooler in (23), free flow of air out into the surrounding environment (*p_a-out_* = 1 bar) was assumed. On the other hand, air pressure at the inlet to the exchanger was *p_a-out_* = *p_a-out_* + Δ*p_a_*.

In order to determine the entropy generation rate in the lamelled air cooler, a calculation algorithm was prepared based on Peclet’s equation and balance equations for the air and the ice slurry [15]. The air-side heat transfer coefficient was calculated from Schmidt’s equation [57], while air-side flow resistance calculations were based on Idelcik’s formula [58]. For the ice slurry, flow resistance values were calculated using Equations (7)–(10). Heat transfer coefficients, in turn, were calculated using Equations (11) and (12).

The calculations were made for:○constant thermal efficiency values $\dot{Q}=$ 20; 40 kW,○constant air flow stream ${\dot{m}}_{a}=$ 2.2 kgs^−1^,○constant inlet air temperature *T_a-in_* = 20 °C,○various ice slurry mass flux values 0.3 $\leq \dot{m}\leq$ 1.25 kgs^−1^,○various mass fractions of ice 5 ≤ *x_s_* ≤ 30%,○constant number of pipes and feeds in the heat exchanger made of *d_i_* = 0.01 m pipes.

The result of the calculations were different heat exchange surfaces (different pipe lengths), which implied different entropy generation rates caused by the flow resistances of the refrigerant and air. A change in the entropy generation rate also resulted from variable mass fluxes of the ice slurry and changes in the specific enthalpy of the ice slurry. Figure 8a shows how the mass fraction of ice affects the entropy generation rate in the air cooler in the event of full melting of the ice slurry (variable mass flux of the ice slurry). Regardless of the thermal efficiency, the minimum entropy generation rate criterion prefers the use of ice slurries with the maximum mass fractions of ice. Low mass fractions of ice generate large ice slurry mass fluxes and high flow velocities (higher than 1 ms^−1^). This results in a significant increase in entropy generation associated with ice slurry and air-side flow resistances. The Bejan number for a 40 kW cooler efficiency changed from 0.04 to 0.43 for the mass fractions of ice of *x_s_* = 30% and *x_s_* = 5%, respectively. Figure 8b shows how the mass share of ice and the mass flux of the ice slurry affect the total entropy generation rates in the air coolers (non-complete ice melting scenario). Regardless of the mass flux, the minimum entropy generation rates were obtained for the maximum mass share of ice x*_s_* = 30%. For a constant mass flux of the ice slurry, the dominant entropy generation component is the entropy generated by the air-side heat transfer the air-side heat transfer process. Lower mass fractions correspond to higher mean temperatures of the ice slurry. Therefore, the receipt of the same heat flux is conditional upon a larger change in the specific enthalpy of the refrigerant. The entropy generation rate as a function of the mass flux is not a monotonic function. This has been illustrated by Figure 9a.

Calculations performed for a constant thermal efficiency of $\dot{Q}=$20 kW and for the mass fraction of ice of *x_s_* = 30% indicate that for the discussed structure of the exchanger, function $\Delta \dot{S}=f(\dot{m},x_{s}=30\%)$ has its minimum value at the point corresponding to the mass flux of $\dot{m}=$0.83 kgs^−1^. At the point of the minimum of function $\Delta \dot{S}$, the Bejan number was 0.2, varying in the entire discussed mass flux range between 0.07 ≤ Be ≤ 0.28. The results presented in Figure 9a refer to various surface areas and therefore also to various heat flux densities. The flow velocity curve makes it possible to determine the ice slurry flow velocity in heat exchanger pipes for particular mass fluxes.

Table 4 shows a comparison of the optimum flow velocities of a 30% ice slurry, determined for flow in a *d_i_* = 0.01 m pipe. Just as in the case of flow through a straight pipe, in the discussed heat exchanger higher heat flux densities also corresponded to higher optimum flow velocities. The comparison (Table 4) indicates that when designing a heat exchanger, it is possible to determine the operational parameters of the cooler which meet the $\dot{S}$~${\dot{S}}_{\min}(d,\dot{q})$ condition by finding the ice slurry flow velocity using Equation (14).

### 3.2. Fluid Cooler

If the same assumptions are made as in the air cooler scenario, and if we take fluid cooling into account, the total entropy generation rate in the cooler can be described using Equation (24):(24)$$\Delta \dot{S}=\Delta {\dot{S}}_{H}+\Delta {\dot{S}}_{\Delta p}=\frac{\dot{m}\;\Delta i_{IS}}{{\bar{T}}_{IS}}+{\dot{m}}_{f}\;c_{pf}\ln\left(\frac{T_{f-out}}{T_{f-in}}\right)-{\bar{\dot{V}}}_{f}\frac{\Delta p_{f}}{{\bar{T}}_{f}}\;-{\bar{\dot{V}}}_{IS}\frac{\Delta p_{IS}}{{\bar{T}}_{IS}}$$

In order to determine the entropy generation rate in a plate heat exchanger/fluid (milk) cooler, an algorithm [59] was developed, where the heat transfer coefficients and flow resistance values on the side of the cooled fluid were calculated on the basis of the equations proposed by Tarasov [60,61]. For the ice slurry, flow resistance values were calculated using Equations (7)–(10). Heat transfer coefficients, in turn, were calculated using Equations (11) and (12), supplemented with coefficients *B_i_* for a rectangular cross-section, included in Table 2.

The calculations were made for:○constant thermal efficiency values $\dot{Q}=$20; 30, 40 kW,○a change in the temperature of the fluid cooled in the exchanger between *T_f−in_* = 12, 20 °C and *T_f−in_* = 2, 5 °C (*T_f_* = 10, 15 K),○various ice slurry mass flux values 0.3$\leq \dot{m}\leq$12 kgs^−1^,○various mass fractions of ice 5 ≤ *x_s_* ≤ 30%,○two geometrical configurations of the exchanger: *n_p_* = 6, *n_s_* = 1; *n_p_* = 12, *n_s_* = 2.

Figure 10a shows how the mass share of ice and the cooled fluid parameters affect the entropy generation rate in a fluid cooler with the efficiency of $\dot{Q}=$ 30 kW, in a scenario involving total melting of the ice slurry (variable ice slurry mass flux). In such a scenario, the dominant component of the entropy generation is the one generated by thermal processes. The Bejan number in the discussed scenario varied between Be = 0.002–0.05. Due to the form of the second component of the sum (24), higher entropy generation rates are characteristic of greater changes in the temperature of the cooled medium and its lower mass flux. Regardless of thermal efficiency and the geometrical configuration of the exchanger, the use of ice slurries with the maximum mass fraction of ice is preferred, as when the complete ice melting condition is assumed, they imply small flow rates of the refrigerant. Figure 10b presents the impact of the mass share of ice and the mass flux of the ice slurry on the total entropy generation rate in a fluid cooler ($\dot{Q}=$ 30 kW, *n_p_* = 6, *n_s_* = 1). In this calculation variant, the constant mass flux condition implies variable values of changes in the ice slurry’s specific enthalpy and various degrees of ice melting. Just as in the case of the air cooler, regardless of the mass flux, minimum entropy generation rates were recorded for the maximum mass share of ice *x_s_* = 30%. For a constant mass flux of the ice slurry, the dominant component of entropy generation is that generated by the ice slurry-side heat transfer process. In a plate heat exchanger, the entropy generation due to flow resistances is determined by the flow of the ice slurry. In the calculations performed by the authors, the flow was of a laminar nature and the greatest flow resistances were generated for an ice slurry with the mass share of ice of *x_s_* = 30%, for which the dynamic plastic viscosity coefficient and limit shear stress were highest. The highest Bejan number of 0.15 corresponded to *x_s_* = 30%, $\dot{m}$ = 1.87 kgs^−1^.

Figure 9b presents the results of calculations of the entropy generation rate for various thermal efficiencies of the exchanger $\dot{Q}=$20, 30, 40 kW (*n_p_* = 12, *n_s_* = 2) and a constant mass share of ice *x_s_* = 30%. Function $\Delta \dot{S}=f(\dot{m},x_{s}=30\%)$ demonstrates a minimum which shifts towards higher mass flux values as thermal efficiency increases. At the point marking the minimum of function $\Delta \dot{S}$, the Bejan number was 0.27, varying in the entire discussed range of mass flux changes between 0.02–0.93. Note that in the calculations performed, the minimum entropy generation rate (due to the value of Re*_K_* = 380−530) was recorded for laminar flow, but for *w* = 1.8−2.3 ms^−1^. These flow velocity values are significantly higher than the velocity values used in typical plate exchangers (*w* < 0.5 ms^−1^). The application of a flow velocity of *w* < 1 ms^−1^ leads to an increase in the entropy generation rate by at least 14%–67%, depending on thermal efficiency.

The curves presented in Figure 9b correspond to a constant efficiency of the exchanger. Therefore, in the case of a melting process, entropy generation minimization is not equivalent to the maximization of exchanger efficiency. A sample relationship between exchanger efficiency and the entropy generation rate was presented in Figure 11a. Figure 11 presents the entropy generation rate for a plate heat exchanger fed with a 30% ice slurry or 15% ethanol. For ice slurry, the exchanger efficiency ($\eta =\dot{Q}/\left({\dot{W}}_{\min}\;\Delta T_{\max}\right)$) is determined by the thermal capacity of the cooled fluid (${\dot{W}}_{\min}={\dot{m}}_{f}c_{pf}$) and hence it has a constant value. For single-phase cooling, exchanger efficiency may be determined by the thermal capacity of the refrigerant or the cooled fluid (${\dot{W}}_{\min}=MIN({\dot{m}}_{et}c_{pet};{\dot{m}}_{f}c_{pf})$). It is worth noting that the use of the minimum entropy generation rate as the only design criterion may, for example, generate high investment costs. Figure 11b presents sample calculations where for low mass fluxes ($\dot{m}\leq 0.53$ kgs^−1^), lower entropy generation rates are recorded in an ethanol-fed exchanger than in an ice slurry-fed exchanger. If ethanol is used, however, the required surface area of the exchanger is at least 60% higher than in the case of an ice slurry fed exchanger.

## 4. Conclusions

Using the results of experimental studies and formula for the calculation of flow resistance and heat transfer coefficient values during the melting of ice slurry under forced convection conditions, an analysis of the entropy generation rates was performed during flow in straight pipes and heat exchangers.

The study determined the area of optimal parameters for ethanol ice slurry (*w, x_s_*) for different tube diameters and heat flux densities for which the entropy flux generated during the melting process in the conditions of forced convection of ice slurry is minimal. The original aspect presented in the paper is the description of the set of optimal parameters for *w, x_s_,* the criteria relationship between the Reynolds number and dimensionless numbers taking into account the effect of geometric and flow parameters on the generated entropy flux.

A new aspect discussed in this paper was the comparison of different strategies for the selection of flow parameters when using ice slurry as a coolant. Three criteria for the selection of flow parameters of ice slurry were compared: Minimum criterion of entropy flux generated by the slurry during flow in a straight channel, minimum criterion of entropy flux generated in a heat exchanger supplied with ice slurry and generalized overall enhancement efficiency criterion. Attention was drawn to the fact that the fulfilment of the criterion ε_Nu−*f*_ > 1 more significantly limits the velocity of ice slurry flow from above than the minimum entropy flux criterion.

Unlike the criterion of minimum entropy flux of ice slurry in a straight channel, the criterion of minimum entropy flux in a heat exchanger always prefers the use of maximum mass fractions of ice. In the first case, high ice mass fractions minimize the entropy flux only for low flow velocities (*w* < 1 ms^−1^).

## Figures and Tables

**Figure 1 entropy-21-00514-f001:**
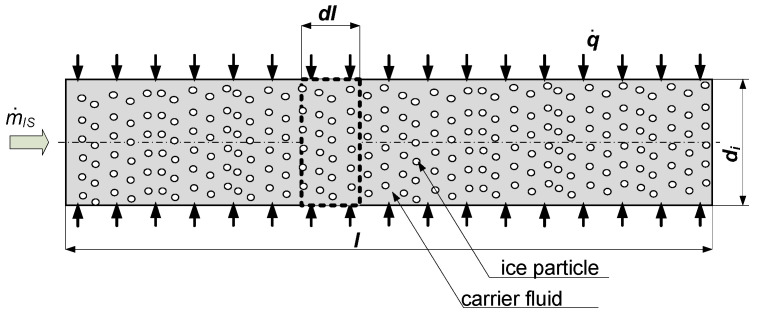
Outline of the heat transfer process in the ice slurry during its flow through a pipe.

**Figure 2 entropy-21-00514-f002:**
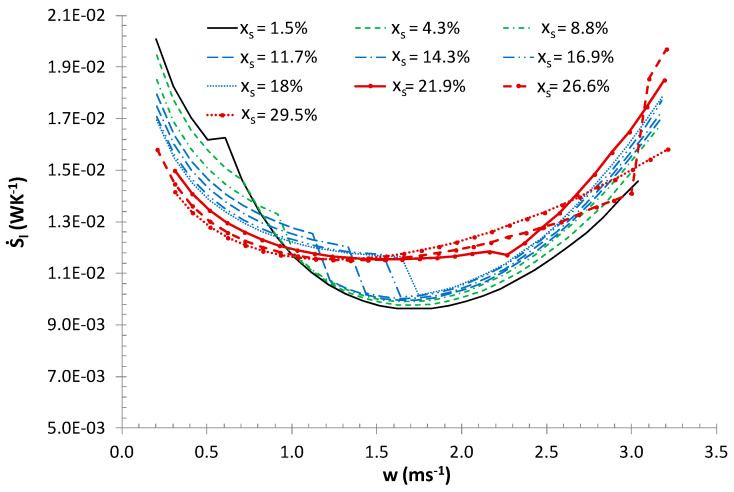
Total entropy generation rate as a function of the mass fraction of ice and ice slurry flow velocity: *d_i_* = 0.016 m, $\dot{q}=8$ kWm^−2^.

**Figure 3 entropy-21-00514-f003:**
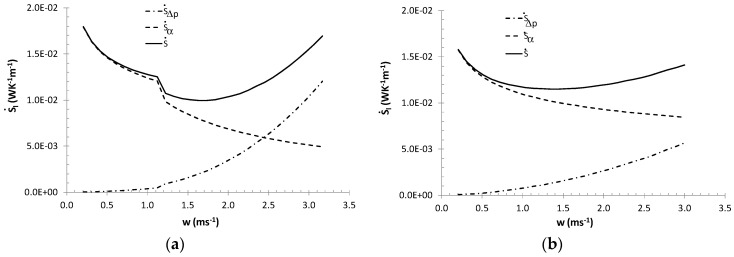
Relationship between entropy generation rates ${\dot{S}}_{H},\;{\dot{S}}_{\Delta p},\;\dot{S}$, for *d_i_* = 0.016 m, $\dot{q}=8$ kWm^−2^: (**a**) *x_s_* = 11.7%; (**b**) *x_s_* = 26.6%.

**Figure 4 entropy-21-00514-f004:**
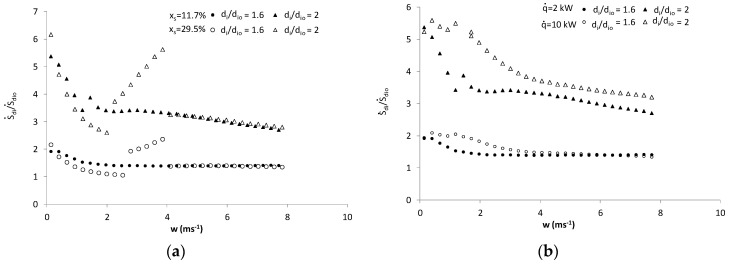
Effect of tube diameter on the generated entropy flux: (**a**) $\dot{q}$= 2 kW, effect of the mass fraction of ice; (**b**) *x_s_ =* 11.7%, effect of heat flux density.

**Figure 5 entropy-21-00514-f005:**
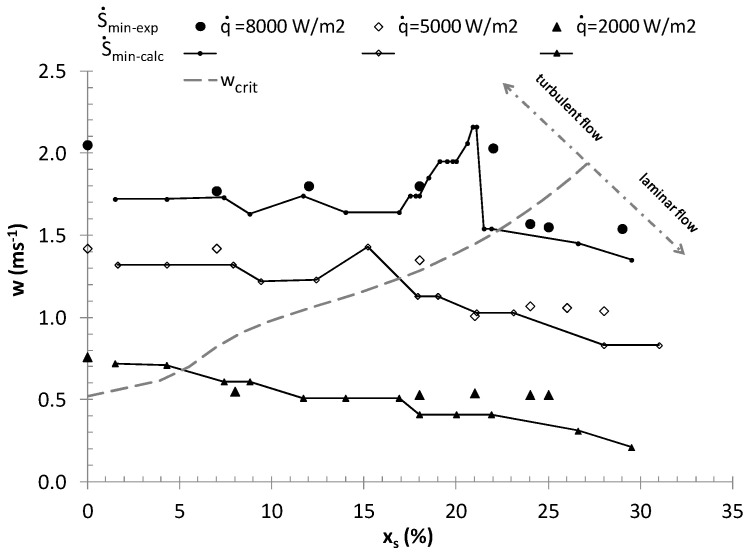
The calculated and experimentally determined parameters *w* and *x_s_*, for which the ice slurry entropy generation rate is at its minimum, *d_i_* = 0.016 m, 2 kWm^−2^
$\leq \dot{q}\leq$ 8 kWm^−2^.

**Figure 6 entropy-21-00514-f006:**
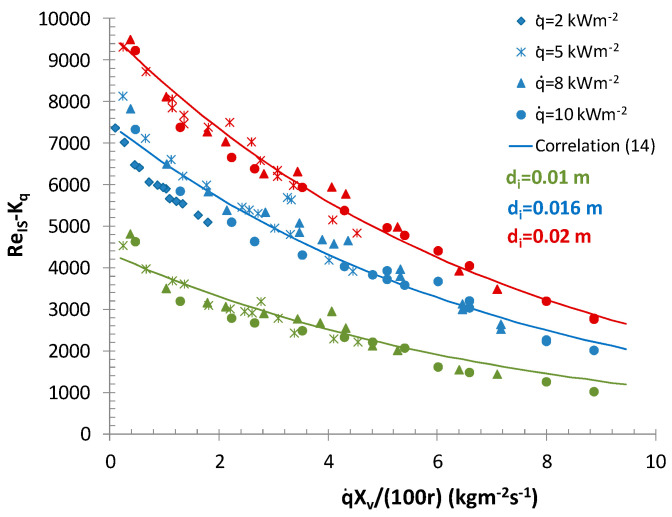
A comparison of the values calculated on the basis of correlation (13) with the determined Re_IS_ and $\dot{q}X_{v}/r$, for which $\dot{S}={\dot{S}}_{\min}$.

**Figure 7 entropy-21-00514-f007:**
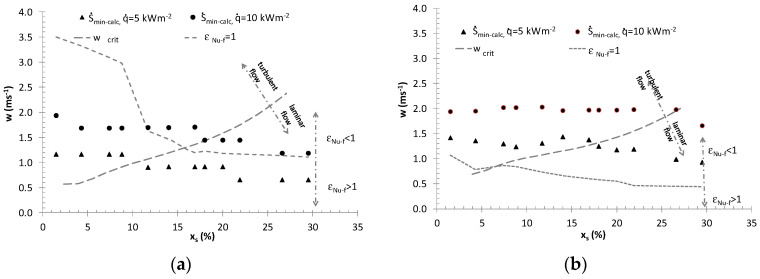
Curves $w{\left(x_{s}\right)}_{\dot{S}\min}$, $w{\left(X_{v}\right)}_{\varepsilon Nu-f=1}$: (**a**) *d_i_* = 0.01 m; (**b**) *d_i_* = 0.02 m.

**Figure 8 entropy-21-00514-f008:**
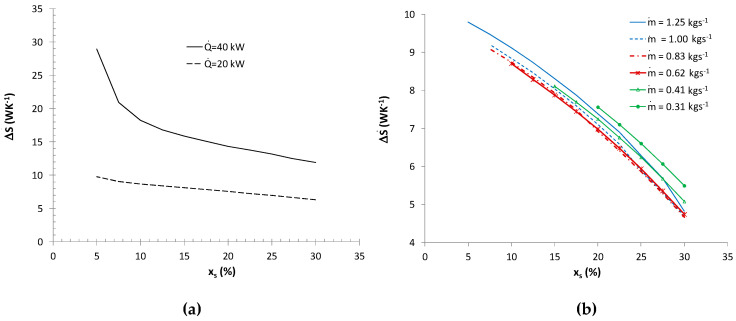
Entropy generation rate in a lamelled air cooler: (**a**) $\Delta \dot{S}(\dot{Q},\;x_{s})$, ${\dot{m}}_{a}=$ 2.2; 3.2 kgs^−1^; (**b**) $\Delta \dot{S}(\dot{m},\;x_{s})$, ${\dot{m}}_{a}=$ 2.2 kgs^−1^, $\dot{Q}$ = 20 kW.

**Figure 9 entropy-21-00514-f009:**
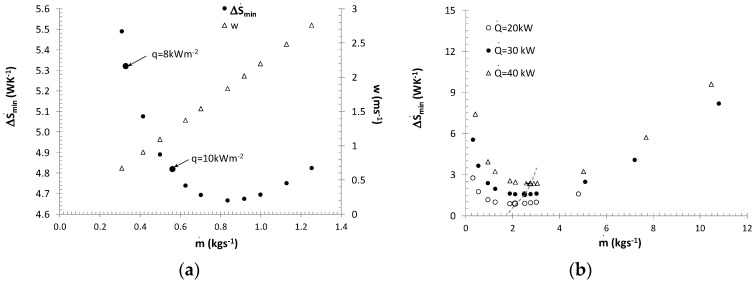
Minimum entropy generation rate $\Delta {\dot{S}}_{\min}(\dot{m})$, *x_s_* = 30%: (**a**) lamelled air cooler $\dot{Q}$ = 20 kW; (**b**) ice water cooler $\dot{Q}$ = 20–40 kW.

**Figure 10 entropy-21-00514-f010:**
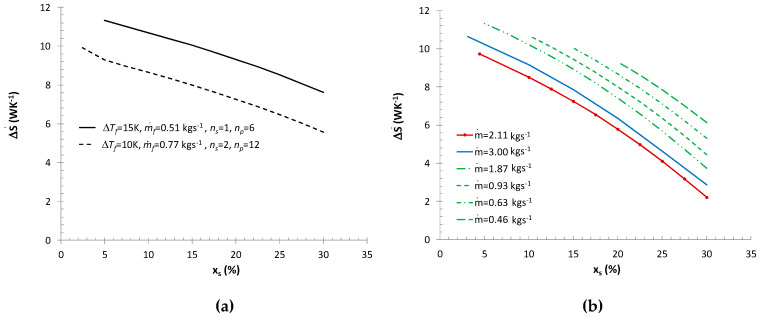
Entropy generation rate in a plate heat exchanger, milk cooler $\dot{Q}$ = 30 kW: (**a**) $\Delta \dot{S}({\dot{m}}_{f},\;x_{s})$, *n_s_* = 1, 2, *n_p_* = 6, 12; (**b**) $\Delta \dot{S}(\dot{m},\;x_{s})$, *n_s_* = 1, *n_p_* = 6.

**Figure 11 entropy-21-00514-f011:**
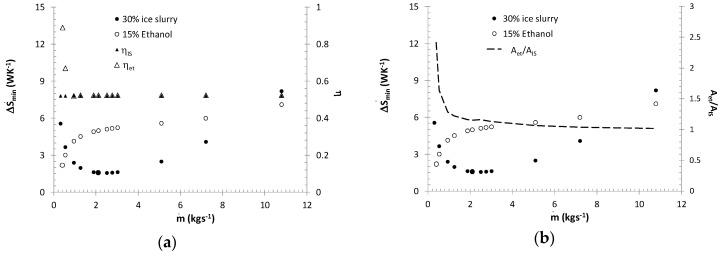
Entropy generation rate in a plate heat exchanger (milk cooler) $\dot{Q}$ = 30 kW fed with a 30% ice slurry or 15% ethanol: (**a**) a comparison of the efficiency *η* of plate exchangers (**b**) a comparison of the surface areas of heat exchangers *A_et_/A_IS_*.

**Table 1 entropy-21-00514-t001:** Equations used to calculate the values of physical properties of the ice slurry.

Properties	Formula	Comments
Enthalpy of ice slurry [51,52]	iIS=(xs100)is+(1−xs100)ia	-
Enthalpy of carrier liquid [51,52]	$i_{a}=\Delta i_{mix}+c_{pa}\left(T-273.15\right)$	*T_ref_* = 273.15 K, *i_a_ = i_a_(x_a_,T)* [51]
Enthalpy of ice [52]	$i_{s}=-r+c_{ps}\left(T-273.15\right)$	*r =* 332.4 kJ kg^−1^
Mean specific heat of ice slurry [51]	cpIS=(xs100)cps+(1−xs100)cpa	*c_pa_* [53], *c_ps_* [51]
Heat conductivity of ice slurry *λ_IS,w=0_* [22]	λIS,w=0=λa⋅[2⋅λa+λs−2⋅(Xv100)(λa−λs)2⋅λa+λs+(Xv100)(λa−λs)]	*λ_a_* [53], *λ_s_* [51]
Ice slurry density	ρIS=(XV100)ρs+(1−XV100)ρa	*ρ_a_* [53], *ρ_s_* [51]
Yield shear stress of ice slurry for *x_ai_ =* 10.6%; *d_s_* = 0.1–0.15 mm [3]	τp=0.013−1.4284(xs100)+73.453(xs100)2−394.64(xs100)3+835.82(xs100)4	-
Plastic viscosity of ice slurry for *x_ai_* = 10.6%; *d_s_* = 0.1–0.15 mm [3]	μp=0.0035+0.0644(xs100)−0.7394(xs100)2+5.6963(xs100)3−19.759(xs100)4+26.732(xs100)5	-
Parameter *K**	$\begin{array}{l} K^{*}={\left(c\mu_{p}\right)}^{n*}\tau_{w}^{1+dn*/c}\\ {\left[\frac{c}{c+d}\tau_{w}^{1+d/c}-\frac{c}{d}\tau_{w}^{d/c}\tau_{p}+\frac{c^{2}}{d\left(c+d\right)}\tau_{p}^{1+d/c}\right]}^{-n*} \end{array}$	*ε_B_=* τ_p_/τ_w_ *c,d* [54]
Parameter *n**	$n^{*}=\frac{c\left[\frac{1-\varepsilon_{B}^{1+d/c}}{c+d}-\frac{\varepsilon_{B}\left(1-\varepsilon_{B}^{d/c}\right)}{d}\right]}{1-\varepsilon_{B}-d\left[\frac{1-\varepsilon_{B}^{1+d/c}}{c+d}-\frac{\varepsilon_{B}\left(1-\varepsilon_{B}^{d/c}\right)}{d}\right]}$	
$X_{v}$	$X_{v}={\left(1+\frac{1-x_{s}}{x_{s}}\frac{\rho_{s}}{\rho_{a}}\right)}^{-1}$	*-*

**Table 2 entropy-21-00514-t002:** Values of coefficients in Equations (11) and (12).

Cross Section	Type of Flow	Parameter	Value	Applicability Range
Pipe $0.01\leq d_{i}\leq 0.02$	laminar	*B* _1_	2.52	3% < *x_s_* < 30% *w_m_* > 0.1 ms^−1^ 200 < Re_K_ < 2100
		*B* _2_	0.11	
		*B* _3_	−0.10	
		*B* _4_	−0.35	
		*B* _5_	0.052	
	turbulent	*B* _1_	0.0096	3% < *x_s_* < 30% *w_m_* <4.5 ms^−1^ 2100 < Re_K_ < 11,000
		*B* _2_	0.70	
		*B* _4_	−0.10	
Rectangular and slit cross-section $0.0055\leq d_{h}\leq 0.012$	laminar	*B* _1_	3.66	5.6% < *x_s_* <3 0% *w_m_* > 0.5 ms^−1^ 30 < Re_K_ < 2300
		*B* _2_	0.16	
		*B* _3_	−0.28	
		*B* _4_	−0.12	
		*B* _5_	0.16	
	turbulent	*B* _1_	0.0032	3% <*x_s_* < 30% *w_m_* < 3.1 ms^−1^ 1900 < Re_K_ < 6000
		*B* _2_	0.86	

**Table 3 entropy-21-00514-t003:** Values of coefficients in Equations (15) and (16).

Parameter	Value
*C* _1_	C1=c11(did0)2+c12(did0)+c13; *c*_11_ = −2292.0231, *c*_12_ = 10,483.068, *c*_13_ = −4879.7335
*C* _2_	C2=c21(did0)2+c22(did0)+c33; *c*_21_ = −0.0708, *c*_22_ = 0.2925, *c*_23_ = 0.6583
*C* _3_	11.3766
*C* _4_	1.15317
*C* _5_	52.0511
${\dot{q}}_{o}$	10,000
${\dot{G}}_{o}$	380
*d_io_*	0.01

**Table 4 entropy-21-00514-t004:** Optimum velocity values according to Equation (14) and for $\Delta {\dot{S}}_{\min}$ from Equation (23) *d_i_* = 0.01, *x_s_* = 30%.

*q*	*w* _tube_	*w_HE_*	Be_tube_	Be*_HE_*
8	0.93	0.72	0.08	0.08
10	1.19	1.23	0.125	0.14

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
