# Peer review of "Entropy Generation of Forced Convection during Melting of Ice Slurry"

_entropy, 2019, doi:10.3390/e21050514_

Round 1

Reviewer 1 Report

Entropy generation of the flow and heat transfer process of ice slurry was investigated in this paper. The influence of mass fraction of ice, slurry flow velocity, heat flux density and pipe diameters were calculated and verified against experimental results. A dimensionless relationship was proposed to determine the interdependency between flow velocity and the volume fraction of ice for which the entropy generation rates were at the minimum. And the entropy generation of ice slurry in heat exchangers were also researched. The paper is interesting, and has no significant errors. However, the novelty of the paper is doubtful. The most creativity part, the dimensionless equation, is lack of basic physical meaning and just like a data fitting. The value of the guidance based on the minimum entropy generation is also weak and need to be discussed more precisely. Just as the author said in the paper, the design of heat exchanger to realize the minimum entropy generation may cause many problems in real applications.

Other minor points are:

1 The introduction is well organized. However, the references are old. More state of the art researches are highly appreciated. The physical explanation should be add (Line 51-56). 

2 The conclusion is colorless and not logical.

3 The figures are difficult to recognized. Please use different color and symbols.

4 The abbreviation PCM need to be introduced.

5 Please chick the typos. 

Line 33  mass f (fraction) of 

Line 70   ca.

Line 73   di

Line 161   the authors

Table 1    ia

Line 297 and Nomenclature  Xv or XV?

6 The literature about virtual entropy generation in heat exchanger measurement is recommended:

ZF Zhang*, C Jiang, YF Zhang, WX Zhou, BF Bai*,Virtual entropy generation (VEG) method in experiment reliability control: Implications for heat exchanger measurement, Applied Thermal Engineering, 2017, 110, 1476-14824.

Author Response

I would like to kindly thank all the reviewers for their suggestions and critical comments, which I have analysed and tried to take into account in the new version of the article

Reviewer I

Comment: The most creativity part, the dimensionless equation, is lack of basic physical meaning and just like a data fitting.

Reply: The answer to this comment is contained in the text in lines 277-290 and 331-338 and by adding figure 4

Comment: The value of the guidance based on the minimum entropy generation is also weak and need to be discussed more precisely.

Reply: The guidance on flow velocity, mass fraction of ice in the context of generating minimal entropy fluxes in the text in lines 391-398 has been supplemented

Comment 1: The introduction is well organized. However, the references are old. More state of the art researches are highly appreciated. The physical explanation should be add (Line 51-56)

Reply: The literature has been reviewed again and Chapter 1 has been changed to include, among others, literature positions such as [20-22] and [46]. The physical explanation of the effect described in lines 51-56, in the current version is contained in lines 52-59.

Comment 2: The conclusion is colorless and not logical.

Reply: The final conclusions have been largely rewritten, lines 600-616

Comment 3: The figures are difficult to recognized. Please use different color and symbols

Reply: Figures 2, 8 and 10 have been corrected as suggested

Comment 4: The abbreviation PCM need to be introduced.

Reply: The text has been reviewed in the light of the above comment. The PCM acronym is used in lines 122, 124 the phrase “Phase Change Materials” is found nowhere in the text, apart from the original titles of the papers. The PCM acronym is explained in the list of designations – line 647

Comment 5: Please chick the typos. 

Reply: Editorial errors have been corrected accordingly, currently in the lines: 29, 73, 76, 151, 662, Table 1, Nomenclature (Appendix A).

Comment 6: The literature about virtual entropy generation in heat exchanger measurement is recommended:

Reply: The relevant publication has been reviewed and added to the references under item no. [46] and included in the text in lines 98 and 116-118.

Reviewer 2 Report

Ice slurry-based refrigeration is one the promising refrigeration technologies which has a variety of applications. It is essential to improve its performance. In the current work, entropy generation approach is applied to characterize the ice slurry flow in straight pipes and typical heat exchanger structures used in refrigeration and air-conditioning technology. The influence of key parameters on the entropy generation characteristics are investigated mainly through analytical method. Generally, the topic of the paper is interesting. However, some problems should be addressed.

1. As can be seen in the introduction part and references, several existing works have been done on the entropy generation involving ice slurries. Comparing with those existing works, what is the main advancement of the current work?

2. The abstract and conclusion part is suggested to be revised to highlight the objective, method, main contribution of the current work.

3. The organization of the part “1. Introduction” is suggested to be improved. It is unnecessary to list many literatures irrelevant to the main topic.

4. In the part “2.3. Results of Calculations”, it is suggested that the author could present detailed information towards Calculations, e.g., conditions, calculation flow, accuracy and validation of the results. 

Author Response

I would like to kindly thank all the reviewers for their suggestions and critical comments, which I have analysed and tried to take into account in the new version of the article

Reviewer II

Comment 1: As can be seen in the introduction part and references, several existing works have been done on the entropy generation involving ice slurries. Comparing with those existing works, what is the main advancement of the current work?

Reply: The literature was reviewed once again in the context of entropy generation in thermal and flow processes with ice slurry. The number of such publications is limited, and their substantive scope is limited or substantially different from the scope undertaken in the presented article. Additional publications concerning the generation of entropy flux in thermal and flow processes with ice slurry were introduced: [40], [41], [47]. The relevant paragraph on this subject is included in lines 119-120 and 127-138. The new aspects discussed in the study are further specified in lines 156-165.

Comment 2: The abstract and conclusion part is suggested to be revised to highlight the objective, method, main contribution of the current work..

Reply: Both chapter Introduction (lines 156-165) and Conclusions (lines 600-6196) were corrected with this comment in mind

Comment 3: The organization of the part “1. Introduction” is suggested to be improved. It is unnecessary to list many literatures irrelevant to the main topic.

Reply: Much of the Introduction chapter has been rewritten, removing the paragraphs that did not contribute key information to the method of calculation or interpretation of the results. Additional publications were introduced in line with the suggestions of the reviewers.

Comment 4: In the part “2.3. Results of Calculations”, it is suggested that the author could present detailed information towards Calculations, e.g., conditions, calculation flow, accuracy and validation of the results. .

Reply: Information clarifying the conditions for calculations is contained in lines 225-228, which are supplemented by information on “forced convection during melting” (line 225) and mass flux of ice slurry 0.04<m_IS<0.62kgs-1 (lines 228).

Validation of the results of entropy flux determination is discussed in lines 303-305, 322-327 and Figure 5. The accuracy of the relationships used in the paper to calculate the heat transfer coefficients and flow resistance was discussed by the author in studies [2], [12], [19], [26], [56] commentary in lines 215-218.

Reviewer 3 Report

Beata Niezgoda-Żelasko has studied the Entropy Generation of Forced Convection During Melting of Ice-Slurry. In my opinion, this work is exciting, but it needs some corrections. My suggestions are as follows:

1-      The abstract is long. Please eliminate unnecessary sentences.

2-      The quantitative results should be mentioned in the abstract.

3-      It would be better if the author avoids using “we” in writing the article.

4-      It would be better for the authors to highlight the state of the art of the study

5-      Adding the recently published literature can improve the introduction. The introduction should be the more elaborate highlighting theist of importance of the subject, very salient previous research work and the need of the present work. Besides, the author can also include the following articles:

-Effects of magnetic field on the convective heat transfer rate and entropy generation of a nanofluid in an inclined square cavity equipped with a conductor fin: Considering the radiation effect.

-Comparison of the effect of five different entrance channel shapes of a micro-channel heat sink in forced convection with application to cooling a supercomputer circuit board.

-Hydrothermal analysis of turbulent boehmite alumina nanofluid flow with different nanoparticle shapes in a minichannel heat exchanger using two-phase mixture model.

-Entropy generation in thermal systems with solid structures – A concise review.

-Entropy generation vs energy efficiency for natural convection based energy flow in enclosures and various applications: A review.

6-      Use the word “respectively” at the end of line 202.

7-      The author should explain about the effect of pipe hydraulic diameter on equations. Does the author consider the impact of hydraulic diameter? How?

8-      What are the boundary conditions? Boundary conditions should be mentioned in more details, for example in a Table or on the figure of the problem statement.

9-      Figure 2 should be improved and presented in color.

10-  The author should explain how he obtained flow resistance and heat transfer coefficient values during the melting of ice slurry under forced experimentally?

11-  What are the experimental setup conditions?

12-  The author must add a picture or at least draw the schematic of the experimental setup

Author Response

I would like to kindly thank all the reviewers for their suggestions and critical comments, which I have analysed and tried to take into account in the new version of the article

Reviewer III

Comment 1: The abstract is long. Please eliminate unnecessary sentences

Reply: The summary has been redacted from 236 to 171 words

Comment 2: The quantitative results should be mentioned in the abstract.

Reply: In the context of this observation, the Conclusions chapter has been revised, in particular lines 600-616

Comment 3: It would be better if the author avoids using “we” in writing the article

Reply: All "autor's own" statements have been eliminated

Comment 4: It would be better for the authors to highlight the state of the art of the study

Reply: The literature was reviewed once again in the context of entropy generation in thermal and flow processes with ice slurry. The number of such publications is limited, and their substantive scope is limited or substantially different from the scope undertaken in the presented article. Additional publications concerning the generation of entropy flux in thermal and flow processes with ice slurry were introduced: [40], [41], [47]. The relevant paragraph on this subject is included in lines 119-120 and 127-138. The new aspects discussed in the study are further specified in lines 156-165.

Comment 5: Adding the recently published literature can improve the introduction. The introduction should be the more elaborate highlighting theist of importance of the subject, very salient previous research work and the need of the present work. Besides, the author can also include the following articles:

Reply: In accordance with the reviewer's suggestion, the literature review was reconstructed and the paper was supplemented with such literature items as [20-22] and [ 40-41], [46-47], and the recommended [24], [48], [49], [51].

Comment 6: Use the word “respectively” at the end of line 202.

Reply: Currently line 202 – changed

Comment 7: The author should explain about the effect of pipe hydraulic diameter on equations. Does the author consider the impact of hydraulic diameter? How?

Reply: The relevant analysis is presented in lines 277-290 and Figure 4.

Comment 8: What are the boundary conditions? Boundary conditions should be mentioned in more details, for example in a Table or on the figure of the problem statement.

Reply: The scope of application of the formulas used for determining the heat transfer coefficients and flow resistances used in the calculations has been included and supplemented in Table 2, commentary in lines 217-218. The scope of calculations presented in chapter 2.3 and their conditions are defined and supplemented in lines 225-228.

The scope of calculations presented in chapter 3.1 and their conditions are defined in lines 435-450.

The scope of calculations presented in chapter 3.2 and their conditions are defined in lines 514-526.

Comment 9: Figure 2 should be improved and presented in color

Reply: The readability of Figure 2 has been improved by adding colours to the curve markings.

Comment 10: The author should explain how he obtained flow resistance and heat transfer coefficient values during the melting of ice slurry under forced experimentally?

Reply: The relationships used in the paper for calculating flow resistance and heat transfer coefficients, dynamic plastic viscosity coefficient and yield shear stress, as well as the criterion used to change the character of motion are original relationships identified by the author, for which appropriate references to literature were used in the paper – [2], [12], [19], [26], [56]. Relevant commentary on the relationships invoked in the study is contained in lines 215-218.

Comment 11: What are the experimental setup conditions?

Reply: As mentioned before, the detailed description of the experimental studies carried out by the author for ethanol ice slurry as well as measurement stations, the course of experiments, measurement accuracy and results were discussed in detail in studies [2], [12], [19], [26], [56]. Because the presented paper concerns the problem of ice slurry entropy flux generation, the paper does not discuss in detail the experimental studies constituting the basis of relationships (7-12) and 13 invoked by the author.

Comment 12: The author must add a picture or at least draw the schematic of the experimental setup

Reply: The reply to Comment 12 is contained in the replies to Comments 10 and 11. Due to the subject matter of the study and its scope, I believe that it is not advisable to supplement it with a description of studies that have already been published by the author. These publications are recognised and quoted by many other authors. In order to convince the reviewer as to the authenticity of the experimental research, I recommend reading the author's publications: [2], [12], [19], [26], [56].

Round 2

Reviewer 1 Report

the revision is accepted

Author Response

Response to the peer reviews

I would like to kindly thank all the reviewer for their suggestions, which I have analysed and tried to take into account in the new version of the article

- If it is necessary, I accept the stylistic changes made by the publisher

Reviewer 2 Report

The problems in the last version were carefully revised, and the main body of the manuscript seems good. However, it is still suggested that the author could revise the abstract,  which seems rather long. In addition, the conclusion part could also be made concise. 

Author Response

Response to the peer reviews

I would like to kindly thank all the reviewer for their suggestions, which I have analysed and tried to take into account in the new version of the article

Reviewer II

Comment: The problems in the last version were carefully revised, and the main body of the manuscript seems good. However, it is still suggested that the author could revise the abstract,  which seems rather long. In addition, the conclusion part could also be made concise. .

Reply: The abstract has been shortened from 172 to 106 words. Lines 10-12 and 15-17 were cancelled.

The summary has been shortened. Lines 617-622 were cancelled.

Reviewer 3 Report

The authors have perfectly addressed all the comments. Now, the paper can be published. 

Good luck. 

Author Response

(The authors gave the same response as above.)
